# Support Vector-based Shapley Value Estimation for Feature Selection and Explanation

## Abstract

In recent years, employing Shapley values to compute feature importance has gained considerable attention. Calculating these values inherently necessitates managing an exponential number of parameters—a challenge commonly mitigated through an additivity assumption coupled with linear regression. This paper proposes a novel approach by modeling supervised learning as a multilinear game, incorporating both direct and interaction effects to establish the requisite values for Shapley value computation. To efficiently handle the exponentially increasing parameters intrinsic to multilinear games, we introduce a support vector machine (SVM)-based method for parameter estimation, its complexity is predominantly contingent on the number of samples due to the implementation of a dual SVM formulation. Additionally, we unveil an optimized dynamic programming algorithm capable of directly computing the Shapley value and interaction index from the dual SVM. Our proposed methodology is versatile, ascertaining feature importance across a myriad of supervised tasks, thereby offering a practical tool for feature selection and explanation. Experiments underscore the competitive efficacy of our proposed methods in terms of feature selection and explanation.

## 1 Introduction

Shapley value-based feature importance, such as Kernel SHAP, is a prominent model-agnostic interpretability method, utilizing an additive feature attribution approach to clarify the predictions of any machine learning model (Lundberg & Lee, 2017; Covert & Lee, 2021). However, its limitations are noteworthy. The additive nature of Kernel SHAP (and many other Shapley valued-based explanations) could lead to oversimplification and loss of nuanced insights, especially in models encoding complex nonlinear relationships and interaction effects. In addition, the precision and reliability of these calculations can be precarious in scenarios featuring high feature correlation or intricate nonlinear interactions, and isolating and quantifying such interactions accurately is challenging. Furthermore, methods like the Shapley-Taylor interaction index (Sundararajan et al., 2020) or Faith-SHAP (Tsai et al., 2023) that do consider interactions rigorously have their own set of challenges, as they involve an exponential number of parameters, complicating both the computation and the interpretation. The assumption of feature independence in Kernel SHAP may lead to unreliable interpretations when there are correlated features, and its reliance on linear regression can introduce approximation errors with highly nonlinear models, besides being computationally intensive with high-dimensional datasets.

In addressing the limitations inherent in existing interpretability models like Kernel SHAP, we propose a method based on the multilinear extension of cooperative games, allowing supervised learning tasks to be modeled as multilinear games. This enables the more comprehensive incorporation and nuanced understanding of interactions between features in contrast to conventional additive models. To tackle the challenge of the exponential growth of parameters intrinsic to models that consider interactions, the dual formulation of support vector machine (SVM) is leveraged. This approach not only serves to circumvent the computational complexity associated with the exponential increase in parameters but also helps provide a representation of the Shapley value based on the dual SVM solution. The methodology promises efficient computation of the newly formulated Shapley value through dynamic programming, contributing a computationally efficient tool for interpreting complex, nonlinear models. The proposed method can be applied to any supervised learning task, so the computed Shapley value could be used both for feature selection and explanation (Section 2).

In summary, the paper has the following contributions: (i) We formulate the supervised learning task as multilinear games, employing direct and interaction effects as the characteristic function of a game; (ii) We effectively handle the exponential parameters in multilinear games by employing SVM, focusing the complexity of the problem on the number of samples rather than the myriad of features and their interactions; (iii) We put forward a data-driven representation of the Shapley value based on the dual SVM solution and devise a dynamic programming algorithm to efficiently compute the Shapley value or interaction index.

**Notation.** We show the matrices with upper-case (unbold-faced) letters, the vectors with bold-faced lower-case, scalars with lower-case (unbold-faced) letters, and the sets with curly upper-case letters. The training set containing $n$ samples is denoted by $\{\boldsymbol{x}_i, y_i\}_{i=1}^n$, where $\boldsymbol{x}_i \in R^d$. The set of all data points $\boldsymbol{x}_i$'s is denoted by $X \in R^{n \times d}$, and $X_{i\bullet}$ and $X_{\bullet j}$ refer to the $i^{th}$ row and $j^{th}$ column of matrix $X$, respectively. The data matrix with all interactions is shown by $\hat{X} \in R^{n \times (2^d-1)}$. We also show the set of features by $\mathcal{F}$ and the set of features with interactions by $\hat{\mathcal{F}}$. Obviously, $\mathcal{F}$ and $\hat{\mathcal{F}}$ contain $d$ and $2^d - 1$ elements, respectively. We also use the one-to-one mapping function $u : \hat{\mathcal{F}} \to Ind$, where $Ind \in [1, 2^d - 1]$ is an integer mapping a set in $\hat{\mathcal{F}}$ to the corresponding column in $\hat{X}$. So, for instance, if we have three features $\mathcal{F} = \{1, 2, 3\}$ and want to have the interaction term $\{1, 2\}$ in the fourth column of $\hat{X}$, then it follows that $u(\{1, 2\}) = 4$. We also use the inverse of $u$ for mapping the index to an interaction set, e.g., $u^{-1}(4) = \{1, 2\}$. The element-wise multiplication, or Hadamard product, is also shown by $\otimes$.

## 2 SUPERVISED LEARNING AND COOPERATIVE GAME THEORY

**Shapley Value and Mobius Transformation.** A cooperative game is characterized by specifying a function for each coalition. For a set of player $\mathcal{F}$, the characteristic function $\mu : 2^{\mathcal{F}} \to R$ assigns a value to each subset of players. The function represents the collective payoff of a set of players when forming the coalition. From a supervised learning perspective, the features and the label (or predicted variable) serve as the players and the payoff, respectively.

Given a characteristic function, the Shapley value is a solution concept in cooperative game theory that concerns the attribution of a payout among the involved players (Shapley, 1953). A salient property of the Shapley value is that it is the unique value that fulfills four axioms (efficiency, symmetry, dummy, additivity). The Shapley value computation is based on the marginal contributions of each feature to all feature subsets. Let $\mu(\mathcal{S})$ be the value for a feature subset $\mathcal{S}$, the marginal contributions of feature $i$ to $\mathcal{S}$ is $\mu(\mathcal{S} \cup \{i\}) - \mu(\mathcal{S}), \forall \mathcal{S} \subseteq \mathcal{F} \setminus \{i\}$. In particular, the Shapley value is the weighted average of all marginal contributions and is defined as (Shapley, 1953):

$$\nu_i = \sum_{\mathcal{S} \subseteq \mathcal{F} \setminus \{i\}} \frac{|\mathcal{S}|!(|\mathcal{F}| - |\mathcal{S}| - 1)!}{|\mathcal{F}|!} \left[ \mu(\mathcal{S} \cup \{i\}) - \mu(\mathcal{S}) \right], \tag{1}$$

where $\nu_i$ is the Shapley value of feature $i$. For feature importance of a learning model, the $\mu(\mathcal{S})$ values can be computed by retaining a model for each subset of features (Lipovetsky & Conklin, 2001), but recent approaches use a sampling approximation of equation (1) with no need of having a model for all feature subsets (Lundberg & Lee, 2017; Mitchell et al., 2022; Datta et al., 2016; Štrumbelj & Kononenko, 2014).

A useful representation of the Shapley value is computed by using the Mobius transformation. The set function $\mu$ can be represented as (Shapley, 1953; Grabisch, 1996):

$$\mu(\mathcal{B}) = \sum_{\mathcal{A} \subseteq \mathcal{B}} m_\mu(\mathcal{A}), \quad \forall \mathcal{B} \subseteq \mathcal{F}, \tag{2}$$

and the Mobius transformation $m_\mu$ can be written as follows:

$$m_\mu(\mathcal{A}) = \sum_{\mathcal{B} \subseteq \mathcal{A}} (-1)^{|\mathcal{A} \setminus \mathcal{B}|} \mu(\mathcal{B}).$$

The Shapley value for feature $i$ as in equation (1) can be represented by using the Mobius transformation as (Grabisch, 1996):

$$\nu_i = \sum_{\mathcal{B} \subseteq \mathcal{F} | i \in \mathcal{B}} \frac{1}{|\mathcal{B}|} m_\mu(\mathcal{B}). \tag{3}$$

The idea of averaging over the marginal contributions for computing the feature importance can be extended to compute the Shapley interaction index. In particular, for any $\mathcal{T} \subseteq \mathcal{F}$, we need to compute the marginal contributions $\mu(\mathcal{S} \cup \mathcal{T}) - \mu(\mathcal{S}), \forall \mathcal{S} \subseteq \mathcal{F} \setminus \mathcal{T}$. The Shapley interaction index is then defined as (Grabisch & Roubens, 1999):

$$I_\nu(\mathcal{T}) = \sum_{\mathcal{B} \subseteq \mathcal{F} | \mathcal{T} \subseteq \mathcal{B}} \frac{1}{|\mathcal{B}| - |\mathcal{T}| + 1} m_\mu(\mathcal{B}). \tag{4}$$

**Multilinear Extension of Games.** The multilinear extension of a cooperative game provides a way to extend a discrete game into a continuous setting (Owen, 1972; 1988a). For a traditional cooperative game with a player (or here feature) set $\mathcal{F}$ and a characteristic function $\mu : 2^N \to \mathbb{R}$, the multilinear extension $G : [0,1]^N \to \mathbb{R}$ is defined as follows:

$$G(\boldsymbol{x}) = \sum_{\mathcal{S} \subseteq \mathcal{F}} \mu(\mathcal{S}) \prod_{i \in \mathcal{S}} x_i \prod_{j \in \mathcal{F} \setminus \mathcal{S}} (1 - x_j) \tag{5}$$

Here, $x_i$ is a continuous variable between 0 and 1 that represents the extent to which player $i$ is part of a coalition. This formula allows for fractional coalition memberships and thereby converts the game into a continuous form. Also, it is shown that the integration of the first derivative of equation (5) results in the Shapley value (Owen, 1972). It is also shown that $G$ can be represented by the Mobius transformation of $\mu$ as:

$$G(\boldsymbol{x}) = \sum_{\mathcal{S} \subseteq \mathcal{F}} m_\mu(\mathcal{S}) \prod_{i \in \mathcal{S}} x_i. \tag{6}$$

The extension to this formulation for multichoice games is also provided, which extends the continuous variable $x_i$ beyond the hypercube (Owen, 1988b; Borkotokey et al., 2015).

**Supervised Learning as Multilinear Game.** The multilinear extension of games as in equation (6) provides a compact formulation that could be used in the supervised learning approaches instead of conventional linear models. We first define multilinear game-theoretic learning.

**Definition 1** *A multilinear model for supervised learning is:*

$$y = h\left(b + \sum_{\mathcal{S} \subseteq \mathcal{F} | j = u(\mathcal{S})} m_j \prod_{i \in \mathcal{S}} x_i\right), \quad \boldsymbol{m} \in R^{2^d - 1}, \boldsymbol{x} \in R^d, \tag{7}$$

*where $h$ is a link function, $b$ is a bias term, and $m_j = m_\mu\big(u^{-1}(j)\big)$.*

The multilinear model, delineated in equation (7), expands upon the traditional linear model by incorporating feature interactions, known in the fields of machine learning and statistics. What sets this model apart is the way interaction effects are interpreted through the lens of the multilinear extension. In this framework, direct and interaction effects are construed as the Mobius transformation of the characteristic function corresponding to their respective feature subsets. This interpretation paves the way for the calculation of critical elements in cooperative game theory, such as the Shapley value and the interaction index, enabling a more profound understanding of feature subset interactions within the model. Another game-theoretic interpretation of this model based on the Harsanyi framework (Harsanyi, 1959; 1963) is presented in Appendix A.

The model (7) is general and can be applied to any supervised task, and the computed Shapley value can be interpreted accordingly. In particular, they can be used in:

- **Feature Selection:** Given a training set, we can train a model (e.g., lasso) based on equation (7), obtain $m_j$'s, and compute the Shapley value accordingly. Such Shapley values represent their importance and can be used to select the most informative features.

- **Local Explanation,** that refers to explaining the prediction of an instance, the features in $\boldsymbol{x}$ are a set of simplified explainable features from the original data. For text classification, for instance, an explainable feature represents the existence of a word in the corresponding document. The data set $X$ is also generated by the permutation of the instance under explanation, and the labels are obtained by feeding the generated data to the learned model, and the computed Shapley value delineates the importance of features in that prediction.

The challenge is that the multilinear model contains an exponential number of parameters, thereby having computational burdens. The next section presents efficient algorithms for such computations.

## 3 Support Vector-based Shapley Value Learning

This section presents a support vector-based method for Shapley value estimation. We focus on the SVM for binary classification in this section, and we present the local explainable model with support vector regression in Appendix B. The results, nonetheless, can be generalized to arguably any kernel learning algorithm.

### 3.1 Kernel Support Vector Machine for Multilinear Extension

**Multilinear feature mapping and SVM.** Given a data point $\boldsymbol{x}$, we define a multilinear feature mapping $\phi_{ML} : R^d \to R^{2^d-1}$ that contains all the feature interactions and is defined as,

$$\phi_{ML}(\boldsymbol{x}) = (x_1, ..., x_d, x_1 x_2, ..., x_{d-1} x_d, x_1 x_2 x_3, ..., x_1 x_2 ... x_d). \tag{8}$$

Then, for a training set for binary classification, the SVM seeks to find a hyperplane $\boldsymbol{m}^T \phi_{ML}(\boldsymbol{x}) + b$ by solving the following minimization (Cortes & Vapnik, 1995):

$$\min_{\boldsymbol{m}, b} \frac{1}{2} \|\boldsymbol{m}\|^2 + C \sum_i \max(0, 1 - y_i(\boldsymbol{m}^T \phi_{ML}(\boldsymbol{x}_i) + b)), \tag{9}$$

where $C > 0$ is a trade-off parameter between the loss function and regularization. Since the dimension of feature space is potentially big (exponential here), the SVM provides a dual formulation for minimization (9) as:

$$\min_{\boldsymbol{\alpha}} \frac{1}{2} \sum_{i,j} \alpha_i \alpha_j y_i y_j \phi_{ML}(\boldsymbol{x}_i)^T \phi_{ML}(\boldsymbol{x}_j) - \sum_i \alpha_i \quad s.t. \sum_i \alpha_i y_i = 0, \quad 0 \le \boldsymbol{\alpha} \le C. \tag{10}$$

The interesting property of problem (10) is that it only depends on the number of samples, and not features (or its mapping to a higher dimensional space). Also, if we map $\boldsymbol{x}_i$'s to a higher dimensional space like $\phi_{ML}(.)$, we only need to know the inner product of the points in that space (i.e., $\phi_{ML}(\boldsymbol{x}_i)^T \phi_{ML}(\boldsymbol{x}_j)$) and use minimization (10) for classification. Realizing such an inner product is known as the kernel trick and the corresponding inner product is known as the kernel function.

**Full and q-additive Kernel functions.** To compute the kernel function in $\phi_{ML}$, we consider the case that we account for all interactions, as well as when we restrict the order of interactions.

For all interactions, we show in the following lemma that the inner product of points in the $\phi_{ML}()$ space can be realized very efficiently[1]. See Appendix C for the proof.

**Lemma 1** *The kernel function for points $\boldsymbol{x}$ and $\boldsymbol{z}$ with the multilinear extension is computed as:*

$$k_{ML}(\boldsymbol{x}, \boldsymbol{z}) = \phi_{ML}(\boldsymbol{x})^T \phi_{ML}(\boldsymbol{z}) = -1 + \prod_{i=1}^d (1 + x_i z_i). \tag{11}$$

Lemma 1 provides a linear-time formulation for computing the kernel function for the multilinear extension. However, for some problems, there is some prior knowledge that restricts the order of interactions among features. In addition, some interaction indices fulfill a set of axioms when the interaction order is restricted (Sundararajan et al., 2020; Tsai et al., 2023). As such, we define the q-order Shapley mapping as follows.

---

[1]The kernel function is similar to the ANOVA kernel with a linear base kernel (Durrande et al., 2013; Stitson et al., 1999), with a minor difference of having a -1. we provide the proof for completeness and because it helps understand the dynamic programming for q-additive kernel function presented in the following.

**Definition 2** *The multilinear mapping is said to be q-order additive, or simply q-additive, if maximum q features can interact. The corresponding kernel is called the q-additive kernel.*

The q-additive multilinear mapping $\phi^q_{ML}(\boldsymbol{x})$ has the following form:

$$\phi^q_{ML}(\boldsymbol{x}) = (x_1, ... x_d, x_1 x_2, ..., x_{d-1} x_d, ..., x_1 ... x_q, ..., x_{d-q} x_d). \tag{12}$$

Given the q-additive multilinear mapping, the q-additive multilinear kernel, shown by $k^q_{ML}$, cannot be computed by equation (11). Instead, we formulate the q-additive kernel as a dynamic programming problem, whose recursive formula is as follows:

$$k^{\tilde{q},\tilde{d}}_{ML}(\boldsymbol{x}, \boldsymbol{z}) = \begin{cases} \sum_{i=1}^{\tilde{d}} x_i z_i & \text{if } \tilde{q} = 1 \\ k^{\tilde{d},\tilde{d}}_{ML}(\boldsymbol{x}, \boldsymbol{z}) & \text{if } \tilde{q} > \tilde{d} \\ x_{\tilde{d}} z_{\tilde{d}} \big( k^{\tilde{q}-1,\tilde{d}-1}_{ML}(\boldsymbol{x}, \boldsymbol{z}) \big) + k^{\tilde{q},\tilde{d}-1}_{ML}(\boldsymbol{x}, \boldsymbol{z}) & \text{otherwise.} \end{cases} \tag{13}$$

In equation (13), $k^{\tilde{q},\tilde{d}}_{ML}$ is the $\tilde{q}$-additive kernel for the first $\tilde{d}$ elements, and $\boldsymbol{x}, \boldsymbol{z} \in R^d$, $\tilde{d} \leq d$. We initialize $\tilde{q}$ and $\tilde{d}$ with $q$ (i.e., maximum order of interaction) and $d$ (number of features), and the output of equation (13) is the q-additive multilinear kernel for $\boldsymbol{x}$ and $\boldsymbol{z}$. The first two cases in equation (13) give the solution for cases the interaction is one (i.e., no interaction) and when the interaction order is greater than $\tilde{d}$, respectively, and the last case captures the recursion computation and is based on the first $\tilde{d} - 1$ elements. Appendix D gives the iterative implementation of the dynamic programming approach presented in equation (13).

## 3.2 Shapley Value Computation: A Dual SVM Representation

Using the multilinear mapping and the corresponding kernel in the dual SVM simplifies the computations. However, we need to compute the primal SVM solution $\boldsymbol{m}$ in order to be able to calculate the Shapley value and interaction index. Given the dual SVM solution $\boldsymbol{\alpha}$, the primal solution could also be computed by the following equation (Cortes & Vapnik, 1995):

$$\boldsymbol{m} = \sum_i \alpha_i y_i \phi_{ML}(\boldsymbol{x}_i). \tag{14}$$

The vector $\boldsymbol{m}$ in equation (14), however, has an exponential number of elements in the number of features and its computation is thus time- and memory-consuming for a large number of features. For only 30 features, for instance, $\boldsymbol{m}$ contains more than one billion elements. We now present some formalizations and algorithms to estimate the Shapley value and interaction index based on the dual SVM solution, thereby circumventing the computational burdens. The following theorem provide such a representation (see Appendix E for the proof).

**Theorem 1** *The interaction index in equation* (4) *can be represented based on the dual SVM solution $\boldsymbol{\alpha}$ as:*

$$I_\nu(\mathcal{T}) = \hat{\boldsymbol{\alpha}}^T \left( \left( \otimes_{i \in \mathcal{T}} X_{\bullet i} \right) \otimes \left( \boldsymbol{1} + \sum_{\substack{\mathcal{B} \subseteq \mathcal{F} \backslash \mathcal{T} \\ |B| \leq q - |\mathcal{T}|, j = u(\mathcal{B})}} \frac{1}{|\mathcal{B}| + 1} \hat{X}_{\bullet j} \right) \right), \tag{15}$$

*where $\boldsymbol{1}$ is a vector of one and $\hat{\boldsymbol{\alpha}} = \boldsymbol{\alpha} \otimes \boldsymbol{y}$.*

Given that $I_\nu(\{i\}) = \nu_i$, the Shapley value is a special case of the interaction index and the above formula. Theorem 1 facilitates the computation of the Shapley interaction index based on the dual SVM solution. However, the computation is still intense since the summation on the right-hand side of equation (15) makes the calculation complex. We now present a dynamic programming approach for computing equation (15). First, we define $\Omega_{\mathcal{T}}$ as:

$$\Omega_{\mathcal{T}} = \sum_{\substack{\mathcal{B} \subseteq \mathcal{F} \backslash \mathcal{T} \\ |\mathcal{B}| \leq q - |\mathcal{T}|, j = u(\mathcal{B})}} \frac{1}{|\mathcal{B}| + 1} \hat{X}_{\bullet j}. \tag{16}$$

Computing $\Omega_{\mathcal{T}}$ efficiently leads to an efficient computation for the Shapley value and interaction index as in equation (22). To that end, we proposed a dynamic programming approach whose recursive formula is given as:

$$
\hat{\Omega}_{\mathcal{T}}^{\tilde{q},\tilde{d},o} = \begin{cases} o^{-1}\mathbf{1} + \sum_{\substack{0<j<\tilde{d} \\ j\notin\mathcal{T}}} X_{\bullet j} & \text{if } \tilde{q}=1 \\ \hat{\Omega}_{\mathcal{T}}^{\tilde{d},\tilde{d},o} & \text{if } \tilde{q}>\tilde{d} \\ \hat{\Omega}_{\mathcal{T}}^{\tilde{q},\tilde{d}-1,o} & \text{if } i=\tilde{d} \\ X_{\bullet\tilde{d}} \otimes \left(\hat{\Omega}_{\mathcal{T}}^{\tilde{q}-1,\tilde{d}-1,o+1}\right) + \hat{\Omega}_{\mathcal{T}}^{\tilde{q},\tilde{d}-1,o} & \text{otherwise.} \end{cases}
\tag{17}
$$

In equation (17), $\Omega_{\mathcal{T}}^{\hat{q},\hat{d},o}$ is the $\hat{q}$-additive summation of the first $\hat{d}$ features, and $o$ is a positive integer responsible for generating the fractions in the Shapley value formula. The first case in equation (17) gives the solution when the interaction order is one, the second case is when the interaction order is bigger than the size of features at that step, the third case is a jump over the feature under explanation, and the last case is the main recursion and is based on the first $\hat{d}-1$ features. Then, $\Omega$ could be computed by setting $\Omega = \hat{\Omega}_{\mathcal{T}}^{q-|\mathcal{T}|,d,1}$. The iterative dynamic programming algorithm to compute $\hat{\Omega}$ for the Shapley value is presented in Appendix F.

### 3.3 OVERALL ALGORITHM AND TIME COMPLEXITY

Given a training set $\{X, y\}$, Algorithm 1 summarizes the steps for computing the Shapley values of the features in the training set.

---

**Algorithm 1** Support Vector-based Shapley Value Learning (SVSVL)

---

**Input** $X \in R^{n\times d}, y, q \in N$
Computing the kernel matrix by Algorithm 2 (see Appendix D)
Solving dual SVM and get the solution $\boldsymbol{\alpha}$
$SV \leftarrow zeros(d)$ # array with d zero elements for Shapley values
**for** i=1:d **do**
    $SV[i] =$ Shapley value for feature $i$ by Algorithm 3 (see Appendix F)
**end for**
**Output:** $SV$

---

The proposed approach is very efficient for high-dimensional data. Training an SVM with the multi-linear kernel is computationally as expensive as other kernel functions and its order is of $O(n)$. The number of operations required is $d$ summations and $2d$ multiplication. For the $q-$additive kernel, we require $2qd$ summations and $qd$ multiplications, which is still very efficient given that $q$ is typically a small number. Having constructed the kernel matrix, solving the SVM has a complexity between $O(n^2)$ and $O(n^3)$. For some particular cases (i.e., least-square SVM), the solution is even obtained by solving a linear equation system. The problem with solving the dual SVM, however, is that it is time-consuming if the number of data points is very large. Nonetheless, the complexity is related to the number of data points $n$, and not $d$ (or $2^d$ when interactions are considered).

For computing the $\hat{\Omega}$ of a continuous-valued feature, the complexity of Algorithm 3 is $O(qd)$, and in each iteration, we need $2qdn$ summations and multiplications. This number of operations could be simplified if we only consider the data points with the corresponding $\boldsymbol{\alpha}$ in the dual SVM nonzero (i.e., the so-called support vectors). Let $n_\alpha$ be the number of support vectors, then the number of operations is reduced to $2qdn_\alpha$ summations and multiplications. Then, for computing the Shapley value given $\hat{\Omega}$, we need $n_\alpha$ summations and multiplications. In total, we need $2qdn_\alpha + n_\alpha$ summations and multiplications for each feature, given the dual SVM solution.

## 4 RELATED WORK

**Feature Selection Methods** Feature selection is pivotal in model construction and interpretation and is typically grouped into two approaches as delineated by (Fleuret, 2004). First, *Filter Methods*

assess feature relevance by computing the correlation between features and labels, employing metrics like mutual information and the $\chi^2$ test (Sánchez-Maroño et al., 2007), offering the advantage of being model-independent. In contrast, *Wrapper Methods* rely on the construction and evaluation of a model to determine the optimal feature subset, with Lasso (O'Brien, 2016) being a prominent representative. It enforces coefficient sparsity, ensuring only the most critical features are included. This work utilizes a similar approach, employing SVM to compute feature Shapley values.

**Shapley Value-Based Explanation Methods** Explainable methods are categorized into local and global approaches. *Local methods*, like SHAP (Lundberg & Lee, 2017), focus on explaining individual instance predictions. Although initially local, SHAP has inspired various other Shapley value-based methods, like *BivariateSHAP* (Masoomi et al., 2021), which accounts for pairwise feature interactions, and L2X (Chen et al., 2018a) and C-Shapley (Chen et al., 2018b), which utilize mutual information to capture feature interactions up to a specific order.

For a *global perspective*, SAGE (Covert et al., 2020) provides model-wide explanations using a permutation-based sampling technique, attributing importance based on an additive assumption. Similarly, ShapleyEffect (Song et al., 2016) employs Shapley values for global explanations, incorporating interaction terms and using Monte Carlo approximation for Shapley value estimation.

*Interaction Indices*, like Shapley-Taylor (Sundararajan et al., 2020) and Faith-SHAP (Tsai et al., 2023), present novel means of interpreting feature interactions using robust axioms. Notably, Faith-SHAP introduces an innovative optimization model closely resembling the multilinear model proposed herein, although its efficiency diminishes with an exponential increase in parameters, necessitating restriction in interaction order.

## 5 EXPERIMENTS

This section presents some experiments comparing the proposed method for use in explanations and feature selection. The implementation is publicly available[2]. All experiments are conducted on a MacBook, with a CPU of 2.3 GHz 8-Core Intel Core i9 and 16GB of RAM.

### 5.1 SYNTHESIZED DATASETS

To do an objective comparison, we construct three synthetic datasets, each embedded with a known ground truth about feature importance. The goal is to contrast the ground truth feature importance with the important features identified by the tested methods. Each dataset contains 10 features. The features are generated according to a normal distribution or a random binary generator. For each synthetic dataset, the target variable is constructed as a known function of a subset of the generated features, making it possible to assess the precision of the feature importance retrieval by the methods. For the first data set, the target variable is assigned as $y \propto X_1 * X_2 * X_3$. For the second data set, we assume more complex interactions among features and set the target variable to $y \propto X_1 * X_2 * X_3 + X_4 * X_5$. For the third one, we assume that the interactions are among the second moments of the first three features, and set the target variable to $y \propto exp\left(\sum_{i=1}^{4} X_i^2\right)$.

For feature selection, we contrast our proposed method, SVSVL, with several established methodologies: Mutual Information (MI) (Fleuret, 2004), K-Best selection utilizing the ANOVA F-value, Recursive Elimination (RE) (Chen & Jeong, 2007), Random Forest (RF) (Breiman, 2001), and Lasso (O'Brien, 2016). We utilize the implementations provided by *scikit-learn* (Pedregosa et al., 2011) for all comparative methods. Each synthetic dataset undergoes feature selection processes, with feature ranks determined by each method. Given our a priori knowledge of the important features in each dataset, we are able to compute and compare the average ranks of significant features derived by different methods. This experimentation cycle, encompassing both data generation and feature ranking, is replicated 100 times for each dataset. The first row of Figure 1 displays the box plot of the average ranks of pivotal features. Considering the first dataset (left panel), where the optimal average rank is 2, methods like K-Best, RE, and Lasso exhibit suboptimal performance in identifying the most critical features. Conversely, RF, MI, and SVSVL exhibit reliable discernment of crucial features, owing to their capability to account for feature interactions, a capability inherently lacking in linear methods like Lasso. Transitioning to the second dataset (middle panel), where interactions

---

[2]It is enclosed as supplementary material but will be uploaded in a repository upon acceptance.

amongst features escalate in complexity, MI struggles to discern the most critical features, whereas RF and SVSVL continue to exhibit proficiency in identifying them. The last scenario focuses on interactions among the higher moments of the features (right panel). Here, SVSVL faces challenges in reliably pinpointing the most vital features due to its inclination to interpret interactions among the features' first moments. In contrast, RF emerges as a robust methodology, demonstrating commendable performance in interpreting interactions among features' higher moments.

We employed the three synthetic datasets to assess the efficacy of our proposed method relative to established explainability methods, namely LIME (Ribeiro et al., 2016), SHAP (Lundberg & Lee, 2017), L2X (Chen et al., 2018a), and BivariateSHAP (Masoomi et al., 2021). Notably, L2X and BivariateSHAP are capable of accommodating feature interactions. We adhered to default settings for all the compared methods. For L2X, we employed the neural architectures tailor-made for their experiments. The first row of Figure 1 presents a box plot detailing the average rank of the retrieved important features

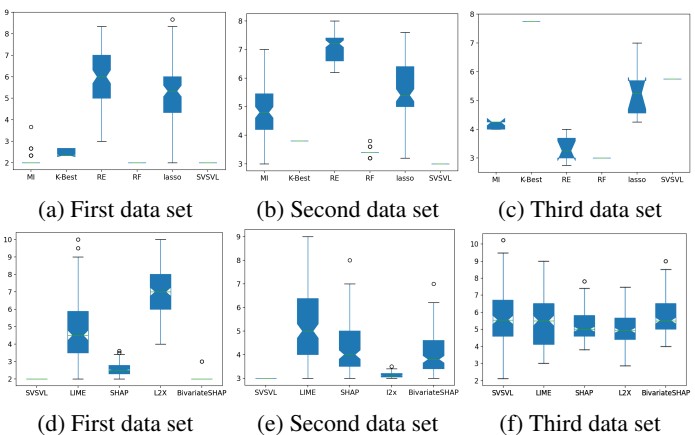

(a) First data set  (b) Second data set  (c) Third data set

(d) First data set  (e) Second data set  (f) Third data set

Figure 1: Comparison of explainable (top row) and feature selection methods (bottom row) on synthesized data sets.

by each method for each dataset; the lower the average rank of the important features is, the better the method. A clear observation is that LIME and SHAP struggle to identify important features, delineated by a higher average rank of the important features. SVSVL, in contrast, exhibits robust performance as a local explainer in the first two scenarios. However, as anticipated, it finds the last scenario more challenging due to the complexity of interactions but still shows superior performance compared to the feature selection scenarios by generating more samples for local explanations. BivariateSHAP also performs well when interactions are not intricate (left panel), given its ability to account only for pairwise interactions. L2X, meanwhile, maintains competitive performance across all cases but faces challenges and slight decreases in performance when interactions intensify in complexity (left panel).

## 5.2 EXPLANATION ON REAL DATA SETS

In this section, we scrutinize the effectiveness and applicability of SVSVL in real-world scenarios, particularly focusing on its comparative performance in local fidelity and execution time against established explainable methods. Our comparative study involves two models: a Bidirectional LSTM model designed for sentiment analysis on IMDB reviews and a Random Forest model, comprising 50 trees, honed on the Boston housing dataset. SVSVL is methodically compared against LIME (Ribeiro et al., 2016), Kernel SHAP (Lundberg & Lee, 2017), L2X (Chen et al., 2018a), and BivariateSHAP (Masoomi et al., 2021). We present some examples of explanation in Appendix G.

We use two metrics for comparison: the execution time and the fidelity score, which is the difference between the prediction of the original model with that of the local surrogate model. We use the mean square error (MSE) to gauge the fidelity of an explainable model. Table 1 shows the MSE between local explainers and the original predictions of the corresponding models, as well as the average execution time for an explanation across 100 different experiments. The discernible trends from the table underscore a conspicuous superiority of models, like SVSVL, that incorporate interactions among features, exhibiting significantly reduced MSE in surrogate models, especially evident in the LSTM model. This enhancement in local fidelity is attributable to their nonlinear nature, enabling a more nuanced replication of the behaviors of the models under explanation. Interestingly, the differences in explaining the Random Forest model were comparatively marginal, reflecting the inherent simplicity of the model where linear methods can be nearly

as effective as their nonlinear counterparts. Within this context, SVSVL stands out by achieving the lowest MSE, highlighting its ability to account for all possible interactions among features.

Delving into the execution time reveals another layer of competitive advantage for SVSVL. It showcases a time efficiency comparable to LIME, which does not account for interactions, and remarkably outperforms SHAP, even with the incorporation of sub-sampling to alleviate computational burdens. The proximity of SVSVL's average execution time to that of LIME further amplifies its competitive stance in the domain of explainable AI. Meanwhile,

Table 1: The comparison of explainable methods based on fidelity of the local explanation and the average execution time in seconds (rounded to the first integer).

| Method | MSE | | Average time (s) | |
|---|---|---|---|---|
| | LSTM | RF | LSTM | RF |
| LIME | $0.12 \pm 0.02$ | $0.003e \pm 0.0$ | 123 | 2 |
| SHAP | $0.11 \pm 0.03$ | $0.004 \pm 0.0$ | 310 | 29 |
| L2X | $0.10 \pm 0.01$ | $0.002 \pm 0.01$ | 73 | 2 |
| BivariateSHAP | $0.07 \pm 0.02$ | $0.004 \pm 0.01$ | 120 | 12 |
| SVSVL | $0.05 \pm 0.01$ | $0.004 \pm 0.0$ | 129 | 3 |

L2X exhibits impressive speed, attributed to its single training stage for providing explanations across the dataset, emphasizing the diverse range of execution efficiencies within the examined explainable methods. In conclusion, the amalgamation of superior local fidelity and competitive execution time positions SVSVL as a robust candidate in the realm of explainable methods, especially when intricate interactions are pivotal.

## 5.3 Time Comparison

We extend our comparison to include an examination of the execution times of the different explainability methods, incorporating Shapley-Taylor and Faith-SHAP into our evaluation. Data sets are randomly generated, each containing between 10 and 20 features, with a known target function. These datasets are then subjected to each explainability method to elucidate 10 randomly selected samples. Figure 2 illustrates the execution times of the various methods corresponding to each feature number. Our analysis was limited to datasets with no more than 20 features due to the prohibitive computational expense of calculating explana-

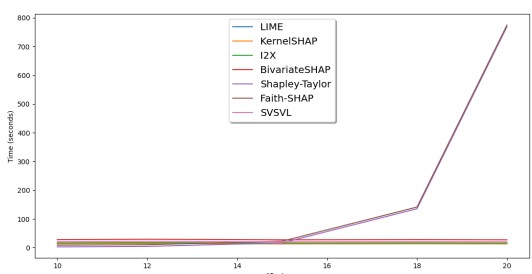

Figure 2: The time comparison of methods.

tions using Shapley-Taylor and Faith-SHAP. All tested methods exhibited roughly equivalent time complexities. As the number of features increases, the execution times of Shapley-Taylor and Faith-SHAP experience a substantial increase. In contrast, the execution times of the other methods tend to increase almost linearly with the increment in the number of features. The disparate trends in execution times highlight the enhanced computational efficiency of SVSVL than Shapley-Taylor and Faith-SHAP, particularly as these methods include the interactions among features as well.

## 6 Conclusion and Discussion

This paper has introduced a novel, non-additive method for computing the Shapley value and interaction index within supervised learning settings, utilizing a multilinear game model, support vector machine (SVM), and efficient dynamic programming algorithms. The key contributions of this work are the multilinear kernel for SVM which considers all feature interactions, a generalized non-additive explanation model applicable to various learning tasks, and a novel representation for the Shapley value based on dual SVM with expedited computational methods, proving effective in feature selection and explanation tasks. However, the method has its limitations, including its inability to account for higher moments of features. A noteworthy consideration is the method's compatibility with $L_2$ regularization in SVMs; it is not applicable to the $L_1$ regularization as the kernel trick cannot be applied directly, preventing full utilization of the method's advantages. Future work may focus on addressing these limitations and exploring alternative interaction functions and further efficient algorithms to identify crucial feature interactions based on the dual SVM.

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

## A  INTERPRETATION OF THE DIRECT AND INTERACTION EFFECTS BASED ON THE HARSANYI DIVIDENDS

In cooperative game theory, the concept of Harsanyi dividends provides a way to allocate the 'marginal contribution' of forming a coalition among its members (Harsanyi, 1959; 1963). Given a characteristic function $\mu : 2^{\mathcal{F}} \to \mathbb{R}$, which assigns a real value to each possible coalition, the Harsanyi dividend $d(\mathcal{S})$ for a coalition $\mathcal{S} \subseteq \mathcal{F}$ is defined as (Harsanyi, 1959; 1963):

$$d(\mathcal{S}) = \mu(\mathcal{S}) - \sum_{\mathcal{T} \subset \mathcal{S}} \mu(\mathcal{T})$$

Intuitively, the Harsanyi dividend $d(\mathcal{S})$ captures the additional value generated by the coalition $\mathcal{S}$ that cannot be accounted for by summing the values of its proper subsets $\mathcal{T}$. In other words, it is the 'extra' benefit realized when the players in $\mathcal{S}$ collaborate, beyond what they could achieve separately in smaller coalitions. Interestingly, if we divide this extra benefit uniformly among the players in the coalition, then the value of each player is tantamount to their Shapley value, i.e.,

$$\nu_i = \sum_{\mathcal{S} \subseteq \mathcal{F}/\{i\}} \frac{d(\mathcal{S})}{|\mathcal{S}|}, \tag{18}$$

which is the same as the Mobius transformation of the Shapley value. So, the $m_j$ in the multilinear model could be construed as the surplus influence of the joint features when they collaborate in a supervised learning task. And, if this surplus is divided equally among the corresponding features, the share of each feature will be tantamount to the Shapley value of the feature.

## B  LOCAL MODEL-AGNOSTIC EXPLANATIONS WITH SUPPORT VECTOR REGRESSION (SVR)

Local explainable methods, like LIME (Ribeiro et al., 2016) and Kernel SHAP (Lundberg & Lee, 2017), aim to explain the predictions of any machine learning model $f : \mathbb{R}^d \to \mathbb{R}$ locally by approximating it with a simpler model $m : \mathbb{R}^d \to \mathbb{R}$. Given an instance $\boldsymbol{x} \in \mathbb{R}^d$, they generate a dataset of perturbed samples $Z = \{\boldsymbol{z}_1, \boldsymbol{z}_2, ..., \boldsymbol{z}'_n\}$ and weighs them using a metric like $\omega$. For LIME, for instance, $\omega$ is defined using a distance function $dis$ as:

$$\omega_i = \omega(\boldsymbol{z}_i) = \exp\left(-\frac{dis(\boldsymbol{x}, \boldsymbol{z}_i)^2}{\sigma^2}\right).$$

The objective is to find an $m$ that minimizes the local loss:

$$\min_{m \in \mathcal{M}} \sum_{i=1}^{n'} \omega_i (f(\boldsymbol{z}_i) - h(\boldsymbol{z}_i))^2 + \Omega(m)$$

where $\Omega$ controls the complexity of the model $m$. Since the local explainable methods are usually concerned with a continuous predicted variable, we need to use support vector regression (SVR) to deal with the continuous predicted variable. For SVR, the primal problem as a local explainer can be written as:

$$\min_{\boldsymbol{m}, b, \xi, \xi^*} \frac{1}{2}||\boldsymbol{m}||^2 + C \sum_{i=1}^{n'} \omega_i(\xi_i + \xi_i^*) \text{ s.t. } \forall i : f(\boldsymbol{z}_i) - (\boldsymbol{m}^T \boldsymbol{z}_i + b) \leq \epsilon + \xi_i, (\boldsymbol{m}^T \boldsymbol{z}_i + b) - f(\boldsymbol{z}_i) \leq \epsilon + \xi_i^*, \xi_i, \xi_i^* \geq 0$$

The corresponding dual problem is:

$$\max_{\alpha, \alpha^*} -\frac{1}{2} \sum_{i,j=1}^{n'} (\alpha_i - \alpha_i^*)(\alpha_j - \alpha_j^*) k(\boldsymbol{z}_i, \boldsymbol{z}_j) - \epsilon \sum_{i=1}^{m} (\alpha_i + \alpha_i^*) + \sum_{i=1}^{m} f(\boldsymbol{z}_i)(\alpha_i - \alpha_i^*)$$

$$\text{subject to: } \sum_{i=1}^{m} (\alpha_i - \alpha_i^*) = 0,\, 0 \leq \alpha_i, \alpha_i^* \leq C\omega_i$$

The learned SVR model $m$ then offers insights into the contribution of each feature locally around the instance $\boldsymbol{x}$. To use the multilinear extension, we add all the interactions to each $z_i$ (i.e., $\phi_{ML}(\boldsymbol{z}_i)$), so algorithms for computing the kernel function for the multilinear extension can be used here as well by employing the dual SVR. In addition, the relationship between the primal and dual solution is:

$$w = \sum_{i=1}^{n'} (\alpha_i - \alpha_i^*)\phi_{ML}(\boldsymbol{z}_i).$$

With some minor adjustments, the dynamic algorithm for the Shapley value computation could also be applied to compute the Shapley value based on the dual SVR problem.

## C  PROOF OF LEMMA 1

We prove equation (11) by induction. For the base case $d = 2$[3], one can write:

$$\begin{aligned}
k_{ML}(\boldsymbol{x}, \boldsymbol{z}) &= x_1 y_1 + x_2 y_2 + x_1 y_1 x_2 y_2 \\
&= x_1 y_1 (1 + x_2 y_2) + x_2 y_2 + 1 - 1 \\
&= -1 + (1 + x_1 y_1)(1 + x_2 y_2),
\end{aligned}$$

which is identical to equation (11) for $d = 2$. We now define arbitrarily $\boldsymbol{x}, \boldsymbol{z} \in R^d$, and also define $\boldsymbol{x}^-, \boldsymbol{z}^- \in R^{d-1}$ by removing the last element in $\boldsymbol{x}$ and $\boldsymbol{z}$, respectively. By induction, we assume that equation (11) holds true for $\boldsymbol{x}^-, \boldsymbol{z}^-$, and prove that it also holds for $\boldsymbol{x}, \boldsymbol{z}$. When the $d^{th}$ element is added to the $\boldsymbol{x}^-, \boldsymbol{z}^-$, two terms are added to the Shapley kernel for the first $d-1$ elements: (1) the multiplication of the $d^{th}$ element (i.e., $x_d z_d$); (2) the terms that $x_d z_d$ create with the previous $d-1$ elements, which is basically the terms in the Shapley kernel for the first $d-1$ elements. Thus, one can write:

$$k_{ML}(\boldsymbol{x}, \boldsymbol{z}) = k_{ML}(\boldsymbol{x}^-, \boldsymbol{z}^-) + x_d z_d + x_d z_d \Big( k_{ML}(\boldsymbol{x}^-, \boldsymbol{z}^-) \Big).$$

The above equation can be rewritten as:

$$\begin{aligned}
k_{ML}(\boldsymbol{x}, \boldsymbol{z}) &= \big(1 + x_d z_d\big) k_{ML}(\boldsymbol{x}^-, \boldsymbol{z}^-) + x_d z_d \\
&= -1 + \Big(1 + x_d z_d\Big)\Big(1 + k_{ML}(\boldsymbol{x}^-, \boldsymbol{z}^-)\Big) \\
&= -1 + \Big(1 + x_d z_d\Big)\Big(\prod_{i=1}^{d-1} 1 + x_i z_i\Big) \\
&= -1 + \prod_{i=1}^{d} \Big(1 + x_i z_i\Big),
\end{aligned}$$

which is identical to equation (11) for $\boldsymbol{x}, \boldsymbol{z} \in R^d$, and that completes the proof.

## D  ITERATIVE DYNAMIC PROGRAMMING FOR Q-ADDITIVE SHAPLEY KERNEL

The recursive formula for computing the q-additive Shapley kernel is discussed in equation (13). We now present the iterative dynamic programming for the q-additive Shapley kernel computation as described in Algorithm 2. Based on this Algorithm, the complexity is of order $O(qd)$. It also requires only $q(2d)$ summations $qd$ multiplications.

---

[3]For $d = 1$, there is no interaction and the equation is obvious.

---

**Algorithm 2** The iterative dynamic programming for computing q-additive Shapley kernel

---

**Input** $\boldsymbol{x}, \boldsymbol{z} \in R^d, q \in N$
$dp \leftarrow zeros(q, d)$ # 2D array with zero elements
$xz = x \otimes z$ # the element-wise product of $\boldsymbol{x}$ and $\boldsymbol{z}$
$dp[0, :] = xz$ # the first row of dp set to $xz$
$sum\_current = sum(xz)$ # sum of all elements in xz
$inner\_prod = sum\_current$ # the result of the inner product
**for** i=1:q **do**
    $temp\_sum = 0$
    **for** j=1:d **do**
        $sum\_current$ -= $dp[i-1, j]$
        $dp[i, j] = xz[j-1] * sum\_current$
        $temp\_sum$+= $dp[i, j]$
    **end for**
    $sum\_current = temp\_sum$
    $inner\_prod$ += $temp\_sum$
**end for**
**Output:** $inner\_prod$

---

## E   PROOF OF THEOREM 1

We assume that $|\mathcal{T}| \leq q$ because otherwise $I_\nu(T) = 0$. We first begin by rewriting the primal SVM solution based on $\boldsymbol{\alpha}$ in equation (14) as:

$$m_j = \hat{\boldsymbol{\alpha}}^T \hat{X}_{\bullet j}, \tag{19}$$

where $\hat{\boldsymbol{\alpha}} = \boldsymbol{\alpha} \otimes \boldsymbol{y}$. Replacing $m_j$ in the Shapley interaction index, one can get:

$$I_\nu(\mathcal{T}) = \sum_{\substack{\mathcal{B} \subseteq \mathcal{F}|\mathcal{T} \subseteq \mathcal{B} \\ |B| \leq q, j = u(\mathcal{B})}} \frac{1}{|\mathcal{B}| - |\mathcal{T}| + 1} m_j = \hat{\boldsymbol{\alpha}}^T \left( \sum_{\substack{\mathcal{B} \subseteq \mathcal{F}|\mathcal{T} \subseteq \mathcal{B} \\ |B| \leq q, j = u(\mathcal{B})}} \frac{1}{|\mathcal{B}| - |\mathcal{T}| + 1} \hat{X}_{\bullet j} \right). \tag{20}$$

The term inside the summation in equation (21) includes the columns of $\hat{X}$ whose index is in $\mathcal{T}$. Thus, one can rewrite the above equation as:

$$I_\nu(\mathcal{T}) = \hat{\boldsymbol{\alpha}}^T \left( \left( \otimes_{i \in \mathcal{T}} X_{\bullet i} \right) \otimes \left( \mathbf{1} + \sum_{\substack{\mathcal{B} \subseteq \mathcal{F} \setminus \mathcal{T} \\ |B| \leq q - |\mathcal{T}|, j = u(\mathcal{B})}} \frac{1}{|\mathcal{B}| + 1} \hat{X}_{\bullet j} \right) \right), \tag{21}$$

and that completes the proof.

## F   ITERATIVE DYNAMIC PROGRAMMING ALGORITHM FOR SHAPLEY VALUE COMPUTATION

Since $I_\nu(\{i\}) = \nu_i$, one can simplify equation (15) as:

$$\nu_i = \hat{\boldsymbol{\alpha}}^T \left( X_{\bullet i} \otimes \left( \mathbf{1} + \sum_{\substack{\mathcal{B} \subseteq \mathcal{F} \setminus \{i\} \\ |B| \leq q - 1, j = u(\mathcal{B})}} \frac{1}{|\mathcal{B}| + 1} \hat{X}_{\bullet j} \right) \right). \tag{22}$$

The recursive dynamic programming for computing $\hat{\Omega}$ with q-additivity constraints is presented in equation (17). The iterative implementation of this algorithm is presented in Algorithm 3. The

---

**Algorithm 3** The iterative dynamic programming for computing $\hat{\Omega}$ used for computing Shapley value

---

$\quad$ **Input** $X \in R^{n \times d}, q \in N$
$\quad dp \leftarrow zeros(q, d, n)$ # 3D array with zero elements
$\quad dp[0, :, :] = X$
$\quad sum\_current = col_sum(X)$ # column-wise summation of the data matrix
$\quad \hat{\Omega} = sum\_current$ # the result of the inner product
$\quad$ **for** i=1:q **do**
$\quad\quad temp\_sum = zeros(n, 1)$
$\quad\quad$ **for** j=1:d **do**
$\quad\quad\quad sum\_current \mathrel{-}= dp[i - 1, j, :]$
$\quad\quad\quad dp[i, j, :] = (i/i + 1) * X[:, j] * sum\_current$
$\quad\quad\quad temp\_sum\mathrel{+}= dp[i, j, :]$
$\quad\quad$ **end for**
$\quad\quad sum\_current = temp\_sum$
$\quad$ **end for**
$\quad$ **Output:** $\hat{\Omega} = sum(dp[:, 0, :])$

---

Table 2: Two examples of the IMDB review data set.

| IMDB examples |
|---|
| Don t waste your time and money on it It s not quite as bad as Adrenalin by the same director but that s not saying much |
| Hated it with all my being Worst movie ever Mentally scarred Help me It was that bad TRUST ME |

complexity of the algorithm is $O(qd)$, and each iteration requires $2qdn$ number of summations and multiplications.

This algorithm could be used to compute the Shapley value and Shapley interaction index.

## G  EXPERIMENTS ON EXPLAINABILITY

We now further show the interactions of words in some of the IMDB reviews. In this regard, consider the two reviews tabulated in Table 2. The top three interactions of each review are also plotted in Figure 3. According to this figure, there is a negative interaction between 'waste', 'bad', and 'not' in the first example, meaning that the collective importance for the three words should be less than the sum of their importance. This is specifically in line with human recognition as the words like 'waste' and 'bad' (and even 'not' for this example) convey a similar sentiment in a review. Also, the interaction detected between 'not', 'much', and 'waster' is interesting as they seem to have a repetitive pattern in the review conveying the same message.

The second example shows positive interactions among 'worst' and 'scarred' as well as among 'worst' and 'hated'. Nonetheless, a negative interaction also exists among the three words, meaning that the existence of three words makes the prediction towards a negative sentiment, while there is a redundancy to consider the three words, and this redundancy is balanced by a positive pairwise interaction.

## H  BROADER IMPACT

This paper presented methodologies for learning, feature selection, and explainability. The selection of features is based on the estimation of the Shapley value, where the feature interactions are included in the form of the multiplications of interacting features. If higher moments of features interact, or interactions are in the form of other nonlinear functions, then the computed Shapley values might not reflect the true importance of the variables, both in feature selection and explanation, which might influence the follow-up decisions made by the proposed method. Also, the merits of other explainable methods for the use case at hand should also be investigated by the users as well. In that regard, the proposed method needs to be evaluated also by humans, and its merits over other

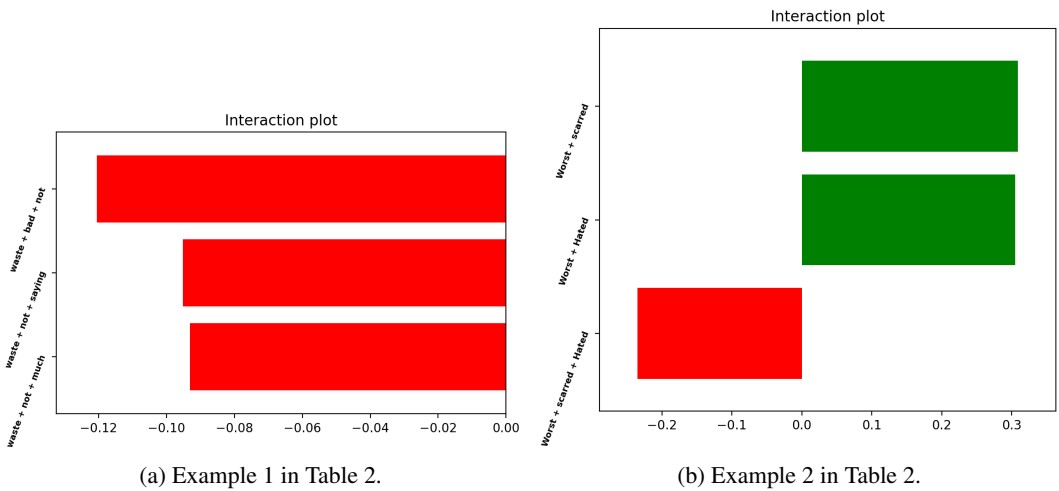

(a) Example 1 in Table 2.    (b) Example 2 in Table 2.

Figure 3: The interaction plots of two examples in Table 2

explainable methods are investigated empirically. Nonetheless, the proposed method can help investigate the biases in a model, find the interactions among features, and assist in applying machine learning to large data sets with feature selection.

