# OpenReview forum: "Support Vector-based Shapley Value Estimation for Feature Selection and Explanation"
_ICLR.cc/2024/Conference — ICLR 2024 Conference Withdrawn Submission_

### Official Review · Reviewer_Ey9T · 2023-10-15

**Soundness:** 2 fair
**Presentation:** 2 fair
**Contribution:** 2 fair
**Rating:** 5
**Confidence:** 4

**Summary:**

The paper proposes a computationally efficient method for assessing feature importance, based on a polynomial model. To this end, it leverages a support vector machine, or more generally the kernel trick. The decisive advantage is added flexibility compared with a linear model.

**Strengths:**

I very much like that the method is able to take all degrees of variable interactions into account. This is a very good use of the kernel trick.

The controlled experiments on synthetic data are well designed.

**Weaknesses:**

Eq. (7) is simply a polynomial hypothesis, possibly feeding into a non-linearity, with degree equal to the number of features. This is an unusual but of course viable model. The authors consider the potential computational burden of the exponential feature explosion, but they do not mention the associated learning theoretical risk, namely overfitting. LASSO-style L1-regularization is hinted at. However, standard L2 regularization is applied in eq. (9), as it is common in SVMs. There is no single word about the need to tune the regularization constant. How do I use a method for estimating feature importance if the resulting values depend on a magic parameter?

Provided that the model is a simple polynomial, and provided that the polynomial kernel is among the few standard kernels considered in basically all SVM research (and all non-linear SVM software packages), I am very much surprised by Lemma 1. Why not simply leverage the polynomial kernel as it is used since 25 years or so?

I am even more surprised by the discussion following eq. (14). For prediction making, an SVM never uses $m$ directly, but only $m^T x$, which can be computed efficiently using the kernel. This is an absolutely basic fact about SVMs and kernel methods in general. I can only conclude that the authors don't really know the methods they aim to leverage. It seems to me that Theorem 1 and the whole section 3.2 can be replaced with the standard prediction making scheme of SVMs. If I am mistaken, then I'd be appreciate being corrected in the rebuttal phase.

For my taste, the paper contains too many references to the appendix for information that is crucial for understanding the proposed method. This essentially amounts to circumventing the page limit. Put differently: ignoring the appending, the paper is not sufficiently self-contained.

The choice of combinations of methods and data sets in section 5.2 appears unsystematic. Why were these data sets used and not others? In order to be convinced, I'd like to see experiments on an established data set collection. In this case, it is fine to present a representative subset, with complete tables in the appendix.

Table 2 presents results in terms of the MSE. Is this simply the quality of the model prediction, i.e., testing the fit of a polynomial to another predictor like a random forest? If so, what does it have to do with estimating the correct feature importance? Also, how do the authors justify its use as a measure of quality of an "explanation"? What if I fit an RF "explainer" to an RF model, with an error of zero? Would the authors then conclude that it is a superior explanation?

Figure 2 can be improved in multiple ways. It definitely needs a log scale on the vertical axis! The range of features from 10 to 20 is extremely narrow, and definitely unsuitable for estimating complexity, which is an inherently asymptotic quantity. Please extend at least to a few hundreds. Simply restrict methods that don't scale well (like the Taylor expansion approach) to smaller dimensions.

I appreciate that the authors provide code. Though, it is not sufficiently commented (or even documented). At the very least, please provide a README explaining prerequisites like required packages, data set files (and where to get them), and an overview of which script is supposed to do what. Naively running the scripts, I was not able to do anything useful with the code, and I was far from reproducing any experiments. Furthermore, the provided zip file contains hidden MACOS and PYTHON temporary files, as well as auto-generated html files and some (huge) javascript. It is entirely unclear to me why they are included.

Minor points:

When introducing the notation, please consider replacing the verb "show" by "denote".

Mobius -> Möbius ({\"o} in LaTex)

Terminology, between eq. (8) and (9): $w^T \phi(x) + b$ is not a hyperplane, but a linear function. For $m \not= 0$, its kernel (zero set) is a hyperplane.

Section 5.1: Please consider replacing the "*" symbol commonly used for multiplication in programming by a LaTeX math symbol like \cdot or \times.

Figure 1: What does the vertical axis represent? It is in the text, but the information should really be in the figure caption.

**Questions:**

Please comment on my criticism above on section 3.2. Can the method make predictions in the same way a kernel SVM does? If no, why not?

Please also comment on my understanding of the MSE in table 2. Why do you think that it is a suitable measure of quality?

---

> ### Author Response · Authors · 2023-11-15
>
> We appreciate the reviewer for providing us with feedback. In what follows, we discuss the raised concerns point-by-point.
>
> 1. Overfitting
>
> Thanks for the nice point. Indeed, tuning the parameter in SVM is important and we will discuss it in more detail in the camera-ready version of the paper. We already tested two hyperparameters (namely: the order of interaction q and the regularization parameter) with the grid search (see the script gridsearch_svsvl.py), but we did not mention it in the manuscript. We can discuss further in the manuscript how they can influence the calculation of the Shapley value.
>
> 2. Polynomial kernel & Lemma 1
>
> The presented kernel in Lemma 1 is different from the polynomial kernel. The polynomial kernel is k_{poly}(x,z) = (x^Tz+c)^r (it is the sum of products of features), while the kernel function in Lemma 1 is k_mul(x,z) = -1 + \prod 1 + x_iz_i (which is the product of products of features).
> The presented kernel function is nonetheless similar to the ANOVA kernel. We already explained this in the text ( cf. Footnote 1, page 4,: “The kernel function is similar to the ANOVA kernel with a linear base kernel (Durrande et al., 2013; Stitson et al., 1999), with a minor difference of having a -1. we provide the proof for completeness and because it helps understand the dynamic programming for q-additive kernel function presented in the following.”) For these two reasons, we provided the result as a lemma.
>
> 3. Prediction with Kernel trick
>
> It is correct that the prediction with SVM can be made by the dual solution and using the kernel trick, and there is no need to find the primal solution – which is not even feasible when the mapping to the feature space is not explicit as in the RBF kernel. However, the goal of Section 3.2 is not to make a prediction but rather to calculate the Shapley values and interaction index for the features. By using the multilinear model discussed in Section 2 in detail, the primal solution to the SVM, denoted by m, has the requisite parameters to compute the Shapley value or interaction indices. That is why we need to compute the primal solution, which is feasible since there is an explicit mapping, i.e., the multilinear mapping. The entire Section 3.2 is devoted to developing a method to compute the Shapley value and interaction index from the dual SVM solution, which will be more efficient given that the primal solution has exponentially many parameters.
>
> 4. Appendices
>
> We tried to have a logical storyline in the paper (even without the appendix), and we moved the proofs, algorithms, and discussions on the interpretation of the multilinear model to the appendix. We also tried to be concise in all sections given the page limit. That being said, we are open to any suggestion to shorten the paper and bring the appendices up in the main part of the text, or to clarify the missed points in the article.
>
> 5. Data sets and methods
>
> We conducted experiments on synthesized and real datasets: with the synthesized experiments we can compare the methods based on the ‘content’ of the explanation, particularly we compare the important features derived by different methods with that of the ground truth. We cannot do the same on the real datasets since the ground truth is not known, and this is why we used the popular measures in explainability (i.e., local fidelity and execution time). We also included several explainable and feature selection methods for the comparative experiments. The used methods are either the most important ones in the literature or the ones that can take into account the interactions of features.

---

> > ### Author Response · Authors · 2023-11-15
> >
> > 6. The mean square error in Table 2
> >
> > For the real data sets, the local fidelity, which is the difference between the prediction of the original model for an instance with that of the local model, is a measure of how well the explainable method is: the better local fidelity indicates that the local method can faithfully imitate the original model in the local neighborhood. This is indeed a common practice in the explainable machine learning realm. For our comparison, we used mean square error (MSE) as a measure of local fidelity.
> >
> > That being said, it is true that the local fidelity does not take into account the explanation ( that is why the synthetic experiments are crucial). To make the comparison more meaningful, we can add the human evaluation for comparison. To do so, we can do the following experiment: for the sentiment classification of the IMDB dataset, we can show to the user the most important words identified by the explainable methods, and we ask the user to guess the sentiment of the review based merely on the explanation (we do not show the entire review). If users get to predict the sentiment of a review correctly based on the provided explanation (which includes the most important words), then the explanations provided to the user are acceptable. Given the time, we can probably attempt to conduct such an evaluation on a few explainable methods (e.g., SVSVL, SHAP, L2X).
> >
> > 7. Improvement of Figure 2 - Time comparison
> >
> > Thanks for the suggestion. We will use the log scale for the time complexity to better compare the explainable methods. Also, we will conduct the comparison on datasets with more features, although the Shapley-Taylor and Faith-SHAP cannot provide an explanation within a reasonable time.
> >
> > 8. The implementation
> >
> > Thanks for the comment. We will indeed add more information (comments and brief documentation) in order for the code to be executed smoothly. The HTML files are for the visualization of explanations, but the MACOS and PYTHON temporary files are indeed redundant and will be removed from the package.
> >
> > 9. Minor points
> > Thanks for reading the article carefully. We apply the mentioned points.

---

### Official Review · Reviewer_iYFx · 2023-10-27

**Soundness:** 2 fair
**Presentation:** 2 fair
**Contribution:** 2 fair
**Rating:** 5
**Confidence:** 4

**Summary:**

In this study, the authors proposed a method for estimating Shapley values (feature importance scores) from the coefficients of polynomial regression.
The authors have derived the relationship between Shapley values and marginal contributions, as well as their Mebius transformations, which connects the coefficients of polynomial regression and Shapley values.
Based on this relationship, the authors propsoed an algorithm for computing Shapley values by learning the coefficients of polynomial regression.
Furthermore, the authors focused on binary classification problems and propose a method to circumvent the direct computation of exponentially many coefficients of polynomial regression through dual SVM.

**Strengths:**

The strength of this paper is on the reduction of the computation of Shapley values into a problem solvable in polynomial time through polynomial regression and dual SVM.

**Originality and Quality**

The use of dual SVM to avoid handling the exponential many coefficients in polynomial regression is the originality of this research.
Furthermore, the computation of Shapley values without explicitly recovering the coefficients of polynomial regression from the dual SVM solution is intriguing.

**Clarity**

Throughout the paper, the main claims were reasonably well described, contributing to overall clarity.

**Significance**

In problems where data or models can be adequately approximated using polynomial regression, the proposed method is considered a valuable approach for computing Shapley values.
Providing an efficient solution for specific class of problems is an important contribution to research in this domain.

**Weaknesses:**

The weakness of this paper lies in the gap between the set function and the polynomial regression model (7).
In the context of Shapley values for feature importance, each input feature's presence or absence is represented by binary values.
The set function is constructed based on this binary representation.
In contrast, each $x_i$ represetns the actual value of each input feature in the polynomial regression model (7).
This usage seems to be inappropriate as an analogy to the set function, as it doesn't correspond well with the notion of presence or absence of features.
In fact, in (7), the condition "a feature takes the minimum value of 0" does not necessarily mean that the feature is absent in the set function.
The paper seems to conflate "presence or absence of features" and "actual values of features" when introducing the multilinear extension.
Thus, Shapley values computed using the "actual values of features" in the polynomial regression model may not align with Shapley values calculated in the original set function.
The validity of replacing "presence or absence of features" with "actual values of features" and its consequences should be discussed in the paper.

Additionally, as a minor weakness, I would like to point out that in Section 5.1, the synthesized datasets seem to have independent features (according to the code in the supplement).
In cases where features are independent, feature importances can be well-estimated by fitting models like RandomForest and calculating permutation importance.
Indeed, in the example below, RandomForest combined with permutation importance ranks important features reasonably well.
(Because I could not find the reproducible codes for Sectoin 5.1, I implemented it by myself.)
The synthesized datasets used in the experiments may be too easy.

```
import numpy as np
from sklearn.ensemble import RandomForestRegressor
from sklearn.inspection import permutation_importance

def data1(X):
    return np.prod(X[:, :3], axis=1), [0, 1, 2]

def data2(X):
    return np.prod(X[:, :3], axis=1) + np.prod(X[:, 3:5], axis=1), [0, 1, 2, 3, 4]

def data3(X):
    return np.exp(np.sum(X[:, :4]**2, axis=1)), [0, 1, 2, 3]

def gen_data(n, datatype, random_state=0):
    np.random.seed(random_state)
    X = np.random.randn(n, 10)
    if datatype == 1:
        y, tif = data1(X)
    elif datatype == 2:
        y, tif = data2(X)
    else:
        y, tif = data3(X)
    return X, y, tif

seed = 0
for dt in range(3):
    x, y, tif = gen_data(500, dt, seed)
    rf = RandomForestRegressor(random_state=seed).fit(x, y)
    r = permutation_importance(rf, x, y, n_repeats=30, random_state=seed)
    print('datatype:', dt)
    print('true important features', tif)
    print('feature ranks', np.argsort(r['importances_mean'])[::-1])

>> datatype: 0
>> true important features [0, 1, 2, 3]
>> feature ranks [2 1 0 3 5 7 9 8 6 4]
>> datatype: 1
>> true important features [0, 1, 2]
>> feature ranks [1 2 0 8 9 6 7 5 3 4]
>> datatype: 2
>> true important features [0, 1, 2, 3, 4]
>> feature ranks [3 4 1 0 2 9 5 8 6 7]

```

**Questions:**

* Are there any justification of replacing "presence or absence of features" in the original set function with "actual values of features" in (7)?
* Is Shapley values computed using the "actual values of features" in the polynomial regression model identical with Shapley values calculated in the original set function?

---

> ### Author Response · Authors · 2023-11-15
>
> We appreciate the reviewer for providing us with feedback. In what follows, we discuss the raised concerns point-by-point.
>
> 1. The relationship between set function and Polynomial Regression, and the presence and absence of a feature
>
> It is correct that the Shapley value-based feature importance is calculated by constructing coalitions between different sets of features by removing (read “absence”) different features. For example, if we have three features $(f_1,f_2,f_3)$, then the coalition $(f_1,f_3)$ is modeled as (1,0,1) and the output from a trained model (for explanation) is calculated as the characteristic function. Having all such coalitions would enable us to calculate the Shapley value, though the value is approximated in practice when we have many features (for the obvious computational burdens).
>
> The main distinction of such an approach with that of the SVSVL is that the latter uses the multilinear extension of games, which transforms the discrete games (as in the presence or absence of a player/feature) into a continuous one. The multilinear extension G of game $\mu$ is defined as ($\mu$ is the characteristic function of a game):
>
> $$
> G(\pmb x) = \sum_{S \subseteq F} \mu(S) \prod_{i \in S} x_i \prod_{j \in F \setminus S} (1 - x_j)
> $$
> The multilinear extension has different interpretations; the probabilistic interpretation (where $x_i \in [0,1]$) entails that each $x_i$ is the probability that $f_i$  joins the coalition. So, if we have $G([0.2, 0.8, 0.9])$ for three features $F = {f_1,f_2,f_3}$, it means that $f_1$  joins the coalition with a probability of 0.2, $f_2$ with a probability of 0.8, and $f_3$ with a probability of 0.9. Thus, $G(x)$ is deemed a generalization of the games where the players can join the coalition with a probability level. A more general interpretation of the multilinear games is developed in multi-choice games, where $x_i$ is not confined to [0,1] (see [1,2]).
>
> Also, it is discussed and proved in Owen 1972 [3] that the Shapley value is a special case of the multilinear extension; in particular, they show that the integration of the derivative of G will lead us to the Shapley value (see Section 2 of the paper). So, computing the parameters of the multilinear extension can be used to compute Shapley value-like feature importance.
>
> That being said, if we get to have binary x’s, as we have for the presence or absence of a feature, then it is easy to prove that $G(x) = \mu(x)$. For example, $G(1,0,1) = \mu({f_1,f_3})$ . So, when constructing coalitions by presence and/or absence of features, G boils down to $\mu$ as expected.
>
> The challenge of such modeling, however, is that there are exponentially many $\mu$ when using multilinear extension, as opposed to the linear regression approximation as in Kernel SHAP.
>
> We have discussed the multilinear extension and the interpretation in Section 2, but we will improve the discussion further to incorporate its meaning when we have binary features (as we have typically in Shapley value-based explanation).
>
> [1] M. Jones et al., Multilinear extensions and values for multichoice games, Mathematical Methods of Operations Research, 2010.
> [2] Borkortoby et al., Fuzzy Bi-cooperative games in multilinear extension form, Fuzzy Sets and System, 2015.
> [3] Owen, Multilinear extension of games, Management Science, 1972.
>
> 2. Independent Features
>
> We distinguish the ‘correlation’ from ‘interaction’: the former indicates how one feature varies with another, while the latter involves the combined effect of two or more features on the target variable.  Given this clarification, it is true that the features for the synthesized experiments are not correlated, but the label y is generated based on the interactions of some features, e.g., $y = X1\times X2\times X3$. We generate such data sets in ‘make_data.py’ for different functions discussed in the experiment section, and the reason for such a choice is to highlight the influence of the interaction in generating explanations.
>
> Also, the random forest can perform very well even when there is strong interaction among features - this is the inherent property of the decision trees. This is also highlighted in our synthesized experiment that includes feature selection with random forest, and the result is plotted in the top row of Figure 1. The code regarding the synthesized experiment is in synthesized_fs.py, which includes the experiment with the random forest as well.

---

> ### Comment · Reviewer_iYFx · 2023-11-23
> **Re: Official Comment by Authors**
>
> > 1. The relationship between set function and Polynomial Regression, and the presence and absence of a feature
>
> I understand that the Shapley value and the multilinear extension are related.
> My point is not on Section 2 but on Section 3.
>
> In Section 2, each $x$ is $\\{0, 1\\}$-valued representing the presence and absence, which is extended to real-valued $x \in [0, 1]$ in multilinear extension.
> Here, even after the extension, $x = 0$ and $x = 1$ still mean that the feature is absent / eixstent.
>
> However, the multilinear feature mapping introduced in Section 3 considers the actual feature values.
> Here, $x = 0$ means that the feature takes the minimum value, and it does not mean that the feature is absent.
>
> Thus, the paper mixes up the two different meanings of $x = 0$, one representing the feature absenece while the other representing that the actual feature value is minimum.
>
> My question is what is the meaning of the multilinear extension (which is justified for $x$ representing absense / eixstense) when combined with the actual feature value.

---

> ### Author Response · Authors · 2023-11-23
>
> We thank the reviewer for the feedback.
>
> We understand that the interpretation of SVSVL in terms of explanation and its relationship with the SHAP method (and similar approaches) is unclear. In what follows we try to shed light on this part by (1) Reviewing the Kernel Shapley additive explanation (Kernel SHAP) and its associated weighted linear regression; (2) Extending the additive explanation by multilinear models and its associated weighted regression; and Finally, (3) how support vector-based estimation can help us deal with the exponentially many parameters in the weighted linear regression in Step (2).
>
> 1. Shapley Additive Explanation (SHAP) and associated weighted linear model
>
> The Shapley value provides a solution concept to attribute the payout among the players in a coalitional game-theoretic setting. For SHAP, the 'payout' is the prediction of the model and the players are the features. In addition, a player can be an individual feature value, e.g. for tabular data. A player can also be a group of feature values. For example, to explain an image, pixels can be grouped to superpixels and the prediction distributed among them [1].
>
> Kernel SHAP takes the following steps to compute the attribution of each explainable feature, and shows that this attribution is the same as the Shapley value:
>
> a) It generates sample coalition $z'_i \in R^{d'}$ ($d'$ is the number of explainable features, e.g., the number of superpixels used to explain an image classifier)
>
> b) Get the prediction of the model under the explanation for the sample $z'_i, i=1,...,n'$; we need to map each sample $z'_i$ to the original space and then get the prediction from the trained model (this is shown in SHAP by h_x(z'_i) ). The prediction on sample z' could be computed based on the expected value with respect to the marginalized distribution.
>
> c) Compute the weight $\omega_i$ for each $z_i$; $$ \omega_i = \frac{d' -1}{{d' \choose |z'|} |z'| (d' - |z'|)}  $$
>
> where $|z'|$ is the number of features present in $z'$.
>
> d) The additive explanation assumes that the prediction (which plays the role of payout) should be distributed among the features as
>
> $ g(z'_i) = b  + \Sigma _{j=1}^{d'} \nu_j z_j$,
>
> where g() is the explainable model, b is the bias term, and $\nu_j$ is the Shapley value of feature j.  The $\nu_j$ is then estimated based on the following weighted linear model as:
>
> $$ \min_{m, b} \Sigma _{i=1}^{n'} \omega_i  (y_i - (b + \nu^Tz'_i) )^2 $$
>
> where $\nu = [\nu_1,...,\nu_{d'}]$ are the Shapley value of $d'$ features.
>
> 2. Multilinear Extension of Games
>
> Instread of using the additive model the associated linear regression, we use the multilinear model and a nonlinear regression. In particular, we assume that the explainable model g is as follows:
>
> $$ g(z') = b + \Sigma_{S \subset F, j = u(S)} m_j \prod_{i \in S} z'_i  = b + m^T \phi _{ML}(z'_i) $$
>
> where we use function u to transform the subset into an index. The vector $\phi _{ML}(z'_i) \in$ {0,1}$^{2^{d'}-1}$ and we have one element for each subset of features. For example, assume we have three features and a coalition that the first and second features are present (i.e., z'=[1,1,0]), and by using $u$ function we have u({1,2}) = 4 and u({1,3}) = 5. Then according to the multlinear extension we have $\phi _{ML}(z') = [1,1,0,1,0,0,0]$, where the first two elements are one because the two features are present, and the fourth element is the element that indicates both features 1 and 2 are present; in fact, the fourth element that corresponds to m({1,2})  in multilinear formulation, that could be written as m({1,2}) $z'_1z'_2$, and since $z'_1z'_2 =1$, we have the value of one in the fourth element. Similarly, the fifth element is zero because the corresponding term in the multilinear extension is m({1,3}) $z'_1z'_3$, and since $z'_1z'_3 = 0$, we have a zero element in the fifth element.
>
> Given the multilinear model, the values $m_j$ could be estimated by the following linear regression (the other parameters like $\omega$ is the same as the one presented above for additive explanation):
>
> $$ min_{\nu} \Sigma _{i=1}^{n'} \omega_i  (y_i - (b + m^T\phi _{ML}(z'_i)) )^2 $$
>
> where $m, \phi _{ML}(z'_i) \in R^{2^{d'}-1}$.  Using the multilinear extension would allow us to interpret the coefficients $m$ in the above regression as the Mobius transformation of $\mu$ in the Shapley value formulation, which allows us to compute the Shaple value accordingly.
>
> [1] Molnar, Christoph. Interpretable machine learning. Lulu. com, 2020.

---

> ### Author Response · Authors · 2023-11-23
> **Minor correction**
>
> 3. Support Vector-based estimation of $m$:
>
> The main challenge in computing the coefficient $m$ in the above formulation is that we have exponentially many parameters. We address this by using the support vector regression and its dual formulation, and computing the Shapley value directly from the dual SVM solution.
>
> The two steps above are briefly explained in Appendix B, and the third step (SVR formulation) is discussed in detail. Upon the acceptance of the paper, we explain the first two steps more clearly.
>
> Minor correction 1: In Section 2, we did not assume that x is binary and we explicitly mentioned that it is continuous! For binary x, the multilinear extension of games converges to the conventional definition of games.
>
> Minor correction 2: The linear regression used in the SHAP also uses the binary features, but the multiplication of the binary features with the coefficient results in the sum of the attribution of the features in the coalition. For example, if $z'=[1,1,0]$, then $z'^T\nu = \nu_1 + \nu_2$, which means that the prediction (or payout) should be distributed among the first two features. We use the same notion in our regression problem, with a major difference of having a coefficient for each subset of features (as in the example discussed in Step (2) above).
>
> [1] Molnar, Christoph. Interpretable machine learning. Lulu. com, 2020.

---

### Official Review · Reviewer_gc9E · 2023-11-02

**Soundness:** 4 excellent
**Presentation:** 3 good
**Contribution:** 3 good
**Rating:** 6
**Confidence:** 3

**Summary:**

This paper introduced a SVM-based method for parameter estimation, and dynamic programing used to compute the shapley value in an efficient way.

**Strengths:**

1. Propose to use SVM to mitigate the inefficiency problem of Shapley values, so that the dynamic programming can be used. This solution is novel.
2. The complexity of the method only relies on data samples.

**Weaknesses:**

1 the advantage of the proposed method is not well presented in this paper, I am not convinced by the effectiveness and efficiency of this method even I have very carefully gone through the related works, introduction, and experimental sections.
2. I strongly encourage the authors to visualize the high-order feature interactions. To see how the methods capture effective feature interactions.
3. Based on the results in section 5.2, it's hard to justify the effectiveness of SVSVL. Maybe the author can show the capability to capture feature interactions.

**Questions:**

1. In fig 2, it looks like most of the methods except Shapley taylor perform similarly. Why the time are not changed with the number of features? How does your method perform better than theirs?

2. Understand that the complexity only relies on the number of samples rather than the number of features. Is it a good property? Generally, the number of data samples is much larger than the number of features, right?

**Details Of Ethics Concerns:**

N.A

---

> ### Author Response · Authors · 2023-11-15
>
> We appreciate the reviewer for providing us with feedback. In the following, we respond to the raised concerns point by point.
>
> 1. The advantage of the method
>
> Thanks for the comment. Except for the related work section – which we devoted entirely to the studies related to our method -  we highlighted the research gap we tried to address, and the advantage of the proposed method based on support vector machine and dynamic programming - See Introduction - Paragraph 2 & 3, Section 3, and the beginning of Section 5. However, we would like to further improve the paper in this respect, if you could kindly point out where you think requires more emphasis on the research gap and the novelty of the proposed method.
>
> 2. Visualizing higher-order feature interactions
>
> Thanks for the suggestion. We admit that it is important to visualize higher-order feature interaction. We do have plots in the appendix highlighting the interactions among features.  In addition, we will add in the final version of the manuscript some multi-level bar charts, each level of which corresponds to the contribution of an order of interaction to the computed Shapley value of a feature.
>
> 3. The advantage of SVSVL - Section 5.2
>
> The advantage of the SVSVL compared to other methods is that it can account for higher-order feature interactions, while the execution time remains the same. We admit that plotting the higher-order feature interactions would give a better intuition of the advantages of the method (we already put some plots in the appendices, but these could be further improved). In essence, keeping the time complexity in order while higher order of interactions are accounted for is the main advantage of the SVSVL. In addition to that, the MSE of SVSVL is lower than other methods, since it uses a nonlinear (yet explainable) model as a local explainer, while other methods like LIME and SHAP use a linear model and have higher MSE as a result (compared to SVSVL).
>
> 4. Advantage of SVSVL - Figure 2
>
> The main point we want to get across by Figure 2 is that the proposed method has very competitive time complexity compared to other additive, linear methods (like SHAP), and is far superior compared to methods that account for higher-order interactions (e.g., Shapley-Taylor and Faith-SHAP). We further improve the figure by plotting the log scale of the execution time (to better compare different methods) and adding datasets with more features (knowing that Shapley-Taylor and Faith-SHAP would not be able to produce results in a reasonable time).
>
> 5. Reliance on the number of samples
>
> The computation of the Shapley value requires an exponential number of parameters in the number of features - the size of the power set of features. So, the improvement we get by using SVSVL is from O(2^d) and O(n3), when d and n are the number of features and samples, respectively. In other words, the fair comparison must be between 2^d and n3.
>
> That being said, it is true that having a large number of samples makes the SVM training quite expensive. This is indeed an old, known problem of kernel methods like SVM and there are some solutions to it, e.g., approximation methods like Nystroem [1]. We will discuss this point further in the Conclusion and Discussion section.
>
> [1] C. Williams & M Seeger, Using the Nyström Method to Speed Up Kernel Machines, NIPS 2000.

---

> > ### Comment · Reviewer_gc9E · 2023-11-23
> >
> > Thanks for the detailed reply, I will slightly adjust my score.

---

### Official Review · Reviewer_NKAm · 2023-11-04

**Soundness:** 3 good
**Presentation:** 2 fair
**Contribution:** 4 excellent
**Rating:** 5
**Confidence:** 3

**Summary:**

This paper proposes an efficient way of computing Shapley value (SV), which is notorious for its combinatorially expensive computational cost. The authors first refer to a known result that the inclusion and exclusion of the variables in SV can be represented as a Mobius transform of a multi-linear form.

Guided by the formal similarity between the multi-linear form and the ANOVA kernel function, the authors propose to use the dual formulation of SVM to compute SV. Thanks to the duality, the dimensionality of the problem is now the number of samples rather than the size of the power set.

 The authors provide theoretical proof of the above conversion and perform comparative empirical studies with alternative attribution methods.

Note: my review is tentative. It may be changed after the discussion period.

**Strengths:**

- Introduced a very innovative view to the SV.
- Provides formal proofs.

**Weaknesses:**

- The description tends to jump directly into the conclusion without showing any intuition.
- The biggest limitation can be that the paper does not provide a direct comparison with the exact SV definition despite the fact that the derived SVM formulation is an approximation, as stated as "the method has its limitations, including its inability to account for higher moments of features".

**Questions:**

- Is the proposed method exact? I mean, does it yield an equivalent attribution value to that from the original definition?
- Please elaborate on what you mean by not being able to account for r higher moments of features, which suggests approximation.
- If it is not exact, direct comparison with the exact definition is mandatory. Did you present such a result in this paper?

---

> ### Author Response · Authors · 2023-11-15
>
> We appreciate the reviewer for providing us with feedback. In the following, we respond to the raised concerns point by point.
>
> 1. The intuition behind the method
>
> The intuition behind the proposed multilinear model is straightforward: If we divide the interaction effect equally among the features involved, the sum of such effects would be tantamount to the Shapley value. For example, if we have two features {1,2}, and we compute m_1, m_2, m_12 as the direct and interaction effects of the two features by, say, linear regression, then the Shapley values of the features are v_1 = m_1 + 0.5 m_12, and v_2 = m_2 + 0.5 m_12 (here m_12 is divided equally between the two features).
> To justify that such a calculation would result in the Shapley value, we provided two theoretical justifications, one based on the multilinear extension and the other based on Harsanyi’s dividends. We explained such an intuition after Definition 1, but if need be, we will clarify this point further in the final version of the manuscript.
>
> 2. Direct comparison with the exact SV
>
> The exact SV in the explainability context means that we calculate the output of the model for each subset of features, and this output plays the role of a characteristic function to compute the Shapley value. For datasets with many features, computing such a function for all subsets of features is computationally expensive, so it is typically estimated by a linear regression model, as in Kernel SHAP for instance.
> We already compared the SVSVL (which is not an exact method) with the Kernel SHAP, but not with the exact method due to the computational burden. A feasible way of comparison to the exact method would be on datasets with a small number of features. We found it more interesting to compare the approximate methods on datasets with larger amounts of features, than using the exact method on too simple datasets. Nonetheless, if need be, we can include further experiments on comparing the SVSVL with the exact methods.
>
> 3. Higher moments of features
>
> The multilinear model, which means the model is linear in each individual feature, cannot account for higher moments of features. For instance, for feature x, there is no inclusion of x^2, x^3,…. Such moments of features are important for learning, but it is not straightforward to include them in the Shapley value or interaction indices calculation, as we require a new set of axioms and a new Shapley-like function for taking into account higher-order moments. We will mention this as a limitation of the proposed method and a venue for future research.

---

> > ### Comment · Reviewer_NKAm · 2023-11-22
> > **In case of Conditional Expectations Shapley?**
> >
> > I'm still not very clear about your thought process showing the equivalence. To be specific, let's use a specific definition of SV based on the expected value with respect to the marginalized distributions (or Conditional Expectations Shapley), which has been adopted in many existing works such as
> > - Molnar, Christoph. Interpretable machine learning. Lulu. com, 2020.
> > - Sundararajan, Mukund, and Amir Najmi. "The many Shapley values for model explanation." International conference on machine learning. PMLR, 2020.
> >
> > In this case, how would you apply your argument in terms of equivalence?

---

> ### Author Response · Authors · 2023-11-23
>
> We thank the reviewer for the feedback.
>
> We understand that the interpretation of SVSVL in terms of explanation and its relationship with the SHAP method (and similar approaches) is unclear. In what follows we try to shed light on this part by (1) Reviewing the Kernel Shapley additive explanation (Kernel SHAP) and its associated weighted linear regression; (2) Extending the additive explanation by multilinear models and its associated weighted regression; and Finally, (3) how support vector-based estimation can help us deal with the exponentially many parameters in the weighted linear regression in Step (2).
>
> 1. Shapley Additive Explanation (SHAP) and associated weighted linear model
>
> The Shapley value provides a solution concept to attribute the payout among the players in a coalitional game-theoretic setting. For SHAP, the 'payout' is the prediction of the model and the players are the features. In addition, a player can be an individual feature value, e.g. for tabular data. A player can also be a group of feature values. For example, to explain an image, pixels can be grouped to superpixels and the prediction distributed among them [1].
>
> Kernel SHAP takes the following steps to compute the attribution of each explainable feature, and shows that this attribution is the same as the Shapley value:
>
> a) It generates sample coalition $z'_i \in R^{d'}$ ($d'$ is the number of explainable features, e.g., the number of superpixels used to explain an image classifier)
>
> b) Get the prediction of the model under the explanation for the sample $z'_i, i=1,...,n'$; we need to map each sample $z'_i$ to the original space and then get the prediction from the trained model (this is shown in SHAP by h_x(z'_i) ). The prediction on sample z' could be computed based on the expected value with respect to the marginalized distribution (as you mentioned) - see equation 9 in [2].
>
> c) Compute the weight $\omega_i$ for each $z_i$; $$ \omega_i = \frac{d' -1}{{d' \choose |z'|} |z'| (d' - |z'|)}  $$
>
> where $|z'|$ is the number of features present in $z'$.
>
> d) The additive explanation assumes that the prediction (which plays the role of payout) should be distributed among the features as
>
> $ g(z'_i) = b  + \Sigma _{j=1}^{d'} \nu_j z_j$,
>
> where g() is the explainable model, b is the bias term, and $\nu_j$ is the Shapley value of feature j.  The $\nu_j$ is then estimated based on the following weighted linear model as:
>
> $$ \min_{m, b} \Sigma _{i=1}^{n'} \omega_i  (y_i - (b + \nu^Tz'_i) )^2 $$
>
> where $\nu = [\nu_1,...,\nu_{d'}]$ are the Shapley value of $d'$ features.
>
> 2. Multilinear Extension of Games
>
> Instread of using the additive model the associated linear regression, we use the multilinear model and a nonlinear regression. In particular, we assume that the explainable model g is as follows:
>
> $$ g(z') = b + \Sigma_{S \subset F, j = u(S)} m_j \prod_{i \in S} z'_i  = b + m^T \phi _{ML}(z'_i) $$
>
> where we use function u to transform the subset into an index. The vector $\phi _{ML}(z'_i) \in$ {0,1}$^{2^{d'}-1}$ and we have one element for each subset of features. For example, assume we have three features and a coalition that the first and second features are present (i.e., z'=[1,1,0]), and by using $u$ function we have u({1,2}) = 4 and u({1,3}) = 5. Then according to the multlinear extension we have $\phi _{ML}(z') = [1,1,0,1,0,0,0]$, where the first two elements are one because the two features are present, and the fourth element is the element that indicates both features 1 and 2 are present; in fact, the fourth element that corresponds to m({1,2})  in multilinear formulation, that could be written as m({1,2}) $z'_1z'_2$, and since $z'_1z'_2 =1$, we have the value of one in the fourth element. Similarly, the fifth element is zero because the corresponding term in the multilinear extension is m({1,3}) $z'_1z'_3$, and since $z'_1z'_3 = 0$, we have a zero element in the fifth element.
>
> Given the multilinear model, the values $m_j$ could be estimated by the following linear regression (the other parameters like $\omega$ is the same as the one presented above for additive explanation):
>
> $$ min_{\nu} \Sigma _{i=1}^{n'} \omega_i  (y_i - (b + m^T\phi _{ML}(z'_i)) )^2 $$
>
> where $m, \phi _{ML}(z'_i) \in R^{2^{d'}-1}$.  Using the multilinear extension would allow us to interpret the coefficients $m$ in the above regression as the Mobius transformation of $\mu$ in the Shapley value formulation, which allows us to compute the Shaple value accordingly.
>
> [1] Molnar, Christoph. Interpretable machine learning. Lulu. com, 2020.
>
> [2] Lundberg and Lee, A Unified Approach to Interpreting Model Predictions, NeurIPS, 2017

---

> > ### Author Response · Authors · 2023-11-23
> >
> > 3. Support Vector-based estimation of $m$:
> >
> > The main challenge in computing the coefficient $m$ in the above formulation is that we have exponentially many parameters. We address this by using the support vector regression and its dual formulation, and computing the Shapley value directly from the dual SVM solution.
> >
> > The two steps above are briefly explained in Appendix B, and the third step (SVR formulation) is discussed in detail. Upon the acceptance of the paper, we explain the first two steps more clearly.
> >
> >
> > [1] Molnar, Christoph. Interpretable machine learning. Lulu. com, 2020.